# The Influence of Deep Eutectic Solvents Extract from Ginger on the Formation of Heterocyclic Amines and Advanced Glycation End Products in Roast Beef Patties

**DOI:** 10.3390/foods11203161

**Published:** 2022-10-11

**Authors:** Yang Xu, Ye Jiao, Jie Luo, Zhiyong He, Maomao Zeng, Qingwu Shen, Jie Chen, Wei Quan

**Affiliations:** 1College of Food Science and Technology, Hunan Agricultural University, Changsha 410128, China; 2School of Food science and Bioengineering, Changsha University of Science & Technology, Changsha 410114, China; 3State Key Laboratory of Food Science and Technology, Jiangnan University, Wuxi 214122, China; 4International Joint Laboratory on Food Safety, Jiangnan University, Wuxi 214122, China

**Keywords:** ginger, deep eutectic solvents, heterocyclic amines, advanced glycation end products, beef patties

## Abstract

Heterocyclic amines (HAs) and advanced glycation end products (AGEs) are important harmful products formed simultaneously during the thermal processing of food. In order to develop a green, efficient method that can be used to control the production of two harmful products simultaneously in food processing. In the present study, deep eutectic solvents (DESs) were used to extract ginger, and this method produced significantly higher levels of total phenolic and flavonoid content as well as an antioxidant activity than ginger extracted using conventional solvents. Herein, we further investigated the inhibitory effects of DES extracts from ginger on the generation of HAs and AGEs in roast beef patties. All the nine DES extracts reduced the formation of HAs and AGEs, and the application of choline chloride–lactic-acid-based DES extract caused a signification reduction of 44.33%, 29.38%, 50.95%, 78.61%, 21.94%, and 17.52% of the PhIP, MeIQx, MeIQ, 4,8-DiMeIQx, Harmane, and Norhamane content, and those for Nε-(carboxymethyl)lysine (CML) and Nε-(carboxyethyl)lysine (CEL) were 49.08% and 58.50%, respectively. Furthermore, the proximate and texture profile changes of beef patties as well as the precursors (creatine, creatinine, and glucose) of HAs and AGEs were evaluated to determine the mechanism of ginger DES extracts on the formation of HAs and AGEs and the physical/chemical changes of ginger DES extracts on beef patties. This study develops a new method for reducing the amount of HAs and AGEs in meat, which will help food manufacturers produce healthier meat products.

## 1. Introduction

With the changing lifestyle around the world, there is an increasing trend in the consumption of high-temperature, processed foods [1]. High-temperature processing can impart unique flavors, colors, and textures to foods as well as increase nutrient digestibility or reduce antinutritional elements [2]. In addition to these beneficial effects, heat treatment can also lead to adverse outcomes, such as the production and accumulation of some harmful substances, including heterocyclic amines (HAs) and advanced glycation end products (AGEs), which have become a research hotspot in the field of food safety in recent years [3,4].

HAs are a class of mutagenic and carcinogenic compounds with various structures formed through Maillard reaction when creatinine, carbohydrates, and amino acids are heated at high temperatures, in which the free radical pathway and the carbonyl pathway are considered to be two main pathways for the formation of HAs [5,6]. Based on the results of epidemiological and animal studies, the IARC have classified most HAs as Class 2A and 2B human carcinogens and recommended limiting the intake of these compounds [7]. AGEs are harmful chemicals formed in nonenzymatic reactions, which refer to the polymerization, condensation, and other reactions between the free amino groups of proteins and the active carbonyls of reducing sugars or intermediate carbonyl compounds of the Maillard reaction [8]. Researchers showed that endogenous AGEs from the daily diet can accumulate in vivo and causing harmful effects in humans, such as increased oxidative stress, inflammation, and the risk of some chronic diseases [8,9].

Given the potential risks and the mechanisms by which HAs and AGEs are formed [10], several effective inhibition strategies such as adjusting cooking conditions have been studied in various food systems to reduce their formation [11]. Since free radicals are believed to be involved in the formation of HAs and AGEs through the Maillard reaction, chemical intervention methods involving the use of various food sources of antioxidants, including spices and herbal extracts, are considered to be promising and effective to inhibit the formation of HAs and AGEs [12,13,14]. Among them, ginger has received increased attention due to its widespread use as a spice and ingredient. It not only can impart special flavor to food but also contains a variety of functional compounds with strong free radical scavenging properties such as polyphenols, flavonoids, and terpenoids [15]. Our previous study found that ginger dose-dependently inhibited HAs and AGEs simultaneously, in which 1.5% of ginger exhibited a strong inhibition capacity against HAs and AGEs (inhibition rates: 27.42% and 35.64%) via their quenching activities of free radical during the Maillard reaction [16]. However, a large amount of spice could adversely affect the flavor and taste of the heat-processed food. Therefore, to improve the inhibition efficiency of natural products against HAs and AGEs in heat-treated foods, researchers have focused on the preparation of natural product extracts with higher antioxidant activities [17,18].

Currently, in food ingredients industries, some conventional organic solvents with strong extraction and dissolving ability are still widely used to extract natural antioxidant compounds. However, since most organic solvents are toxic, volatile, and cause pollution to the environment, their application in the food industry is therefore limited [19]. Deep eutectic solvents (DESs), a novel class of green solvents, are made of elements found in nature that serve as hydrogen bond acceptors (HBAs) or hydrogen bond donors (HBDs) [19,20]. Since DES has the benefits of easy preparation, high purity, no waste generation, and compliance with green chemistry principles, it has already found significant application potential in the extraction of natural ingredients for food fields such as anthocyanins, flavonoids, and isoflavones [20,21].

In the present study, various DESs have been designed and prepared with the ultrasonic bath method, in which choline chloride, betaine, and L-carnitine have been selected as HBA and glycerol, lactic acid, and xylitol as HBD. The aim of the present study to explore how different DES affected the antioxidant capacity of ginger extract and examine how ginger DES extract affected the production of HAs and AGEs in roast beef patties, which provides a theoretical basis for the development of a green, efficient method that can be used to simultaneously control the production of two harmful products in food processing.

## 2. Materials and Methods

### 2.1. Reagents and Chemicals

The ginger powder and raw beef were purchased from local market (Changsha, China), deep eutectic solvents, including choline chloride, L-carnitine, betaine, glycerol, lactic acid, and xylitol, were purchased by Sigma Chemical Co. (St. Louis, MO, USA). Folin–Ciocalteu’s phenol reagent, rutin, gallic acid, 6-hydroxy-2,5,7,8-tetra-methylchroman-2-carboxylic acid (Trolox), 1,1-diphenyl-2-picrylhydrazyl (DPPH), 2,2-azinobis-(3-ethyl-benzthiazoline-6-sulfonate) (ABTS), and 2,4,6-tris-(2-pyridyl)-s-triazine (TPTZ) were purchased from Sigma-Aldrich (St. Louis, MO, USA). AGEs standards, namely CML, CEL, Nε-(1-carboxymethyl)-L-lysine-d4 (CML-d4), and Nε-(1-carboxyethyl)-L-lysine-d4 (CEL-d4), and HA standards, namely PhIP, IQ, MeIQ, MeIQx, 4,8-DiMeIQx, 7,8-DiMeIQx, IQx, Harmane, and Norharmane, were supplied by Santa Cruz Biotechnology, Inc. (Santa Cruz, CA, USA).

### 2.2. Preparation of DES

The previously described heating and stirring procedure was used to prepare DES [15,22]. As donors of hydrogen bonds, the solutions of choline chloride, L-carnitine, and betaine were mixed with acceptors of hydrogen bonds (glycerol, lactic acid, and xylitol) in specific molar ratios and then stirred at 150 rpm and heated at 75 °C to obtain transparent solutions. Table 1 shows comprehensive details on the compositions and molar ratios of the produced DES1-9.

### 2.3. Ultrasonication-Assisted Extraction of Ginger with DES

The ginger powder was combined with a concentration of 75% DES in a 30:1 solvent to solid ratio, and then, each mixture was ultrasonicated for 20 min at 35 °C, 40 kHz, and 600 W by a KQ-300E ultrasonic [15]. After being centrifuged at 1000× *g* for 10 min in a centrifuge, the extract was collected for further testing. Meanwhile, some conventional extraction solvents including water, ethanol, and 80% ethanol solutions (*v*/*v*) were used for comparison.

### 2.4. Determination of Total Phenolic (TPC) and Flavonoid Content (TFC)

The Folin–Ciocalteu technique was used to calculate the TPC [23]. First, 1 mL of Folin–Ciocalteu reagent (0.2 M) and 2 mL of Na_2_CO_3_ solution (75 g/L) were combined with 0.05 mL of each extract. After the combination was reacted at 25 °C in the dark for 2 h, the absorbance of the reacted solutions was measured at 765 nm by a SpectraMax 190 microplate reader. The TPC was calculated and represented as mg GAE/g dw based on the standard curve created with gallic acid (GA).

A colorimetric experiment was conducted to determine the TFC [24]. Then, 2.0 mL samples were mixed with 0.3 mL NaNO_2_ (0.05 g/mL) and incubation for 5 min. Next, 500 μL of AlCl_3_ solution (10%) and 2 mL of NaOH solution (1 mol/L) was added and reacted for 6 min and 10 min at room temperature sequentially. Using a microplate reader, the solution’s absorbance was determined at 510 nm. Rutin was used as the reference substance, and the TFC was calculated based on a calibration curve reacted with rutin (RE) and expressed as mg RE/g dw.

### 2.5. Antioxidant Activity Assays

According to Quan et al. [25], after mixing 20 µL extract solutions with 380 µL of ABTS working solution, the mixture was reacted at room temperature for 10 min. A UV-5300PC spectrophotometer was used to measure the absorbance of the reacting solution at 734 nm.

According to our prior report, the extracts’ ability to scavenge DPPH free radicals was assessed [25]. Briefly, 500 µL samples were mixed with 3 mL of DPPH working solution. A 517 nm absorbance measurement was conducted after the mixtures were incubated at 30 °C for 30 min in the dark.

Based on a previous description by Qie et al., we calculated the ferric ion-reducing antioxidant power (FRAP) [26]. Extract solutions (10 µL) and 190 µL FRAP working solution was reacted for 30 min at 35 °C, and the absorbance of reacted solutions was measured at 593 nm.

Based on Gillespie et al.’s instructions, oxygen radical absorbance capacity (ORAC) assays were conducted [27], and extract solutions were mixed with fluorescein solution (0.2 µM) and 150 mM AAPH (25 µL) and reacted at 37 °C for 10 min. Fluorescence of reacted solutions were read at 485 nm excitation and 530 nm emission.

Based on the standard curve created with Trolox as a standard calibration method, the antioxidant activity of different ginger extracts including ABTS and DPPH free radical scavenging ability, FRAP, and ORAC were calculated and represented as the Trolox equivalent antioxidant capacity.

### 2.6. Meat Preparation and Cooking

Ginger extract from DES and conventional solvents were added to ground beef (with addition levels 0 or 1.0%), and beef patties (40 ± 0.1 g) were prepared using a Petri dish (Φ 6 cm × 1.5 cm) [14,16]. Then, in a SCC61E oven (RATIONAL, Munich, Germany), all patties were roasted at 220 °C for 10 min on each side and stored at −80 °C before further analysis. Beef patties with addition of ginger extract from DES (DES1-9) and conventional solvents (water, ethanol, and 80% ethanol solutions) were named as DES1-9, water, EtOH, and 80EtOH, respectively, while beef patties without addition of ginger extract were defined as control.

### 2.7. Composition, Cooking Loss, and Texture Profile Analysis

According to AOAC procedures [28], the proximate composition of raw and cooked samples, including protein and ash, was analyzed. With the use of a digital pH meter, the pH values of the samples were calculated. Moreover, raw and cooked beef patties were weighed to calculate the cooking loss.

According to our earlier studies, the texture profile of roasted beef patties (1 × 1 × 1 cm) was measured using a TA-XT plus texture analyzer (Godalming, UK) equipped with a P/50 cylinder probe [29]. The parameters for texture profile analysis (TPA) were set as follow: pre-test (4 mm/s), test (3 mm/s), and post-test (4 mm/s) and trigger force (5.0 g with 50% strain for 5 s).

### 2.8. Determination of HAs

We applied our previously reported technique for the extraction of HAs [12,16,29]. In brief, 2 g beef powder, 30 mL NaOH solution (2 M), and 15 mL ethyl acetate were mixed and ultrasonic extracted at 50 kHz, 40 °C, for 40 min. The sample was centrifuged at 3000× *g* for 10 min, and the supernatant was collected for solid-phase extraction [12,16,29].

HAs were analysis by a Waters UPLC-Q-TOF-MS (Milford, USA) with an Acquity UPLC BEH C18 column (50 × 2.1 mm i.d., 1.7 m). A gradient elution method were used to separate the target HAs with a flow rate of 0.3 mL/min, and acetonitrile and ammonium acetate solution (5 mM, pH 6.8) set as mobile phase A and B, respectively. The gradient program was set as: 0 min, 90% B; 10 min, 85% B; 12 min, 0% B; 16 min, 90% B. For the MS analysis, the instrument was set as: capillary voltage (3.0 kV), source temperature (120 °C), desolvation temperature (350 °C), cone gas flow rate (65 L/h), desolvation gas flow rate (650 L/h), and collision gas flow rate (0.15 mL/min). Table 2 shows a summary of the TQD parameters for HAs that were optimized using HA standards.

### 2.9. Determination of AGEs

With a few minor modifications, we used the method we previously described to extract the bound AGEs [12,16]. To remove the lipid, 3 mL n-hexane was added to 30 mg sample and centrifuged for 15 min at 4000× *g*. The defatted samples were next reduced with 3 mL sodium borate buffer and 2 mL NaBH_4_ at 4 °C for 8 h. The mixture was heated at 110 °C for 24 h to facilitate hydrolysis after 5 mL HCl was added. After filtering, solid-phase extraction was performed on 2 mL of redissolved hydrolysates.

The parameters for the UPLC-Q-TOF-MS analysis of AGEs were referred to our previous studies [12,16]. In brief, 5μL of samples were separated in a HSS T3 column (150 × 2.1 mm, 3.5 μm) at 40 °C with gradient elution program: 0–3 min, 1% A; 3–7 min, 3%–100% A; 7–9 min, 100% A; 9–10 min, 100%–1% A, in which acetonitrile and 0.1% formic acid were selected as mobile phase A and B, respectively, and the flow rate was set as 0.3 mL/min. For MS analysis, ESI+ mode and MRM were used for the acquisition of the mass spectrometric data (as shown in Table 2), and the parameters of the MS instrument were the same as in our previous studies [12,16].

### 2.10. Determination of Creatine, Creatinine, and Glucose in Roasted Beef Patties

The content of creatinine and glucose in beef patties was measured by commercial kits using the peroxidase and glucose oxidase method, respectively [17,30]. All experimental operations were performed according to the instructions of commercial kits. According to a method described by Zhang [30], the contents of creatine were measured as following program: 0.5 g samples with 100 mL 30 g/L TCA were homogenized for 1 min and filtered to remove the precipitated protein. Twenty microliters of the extract were defatted with 10 mL diethyl ether. Defatted extract, 2mL diacetyl (0.2 g/L), and 2 mL L-naphthol (25 g/L) were mixed and heated at 50 °C for 6 min, and then, the absorbance was measured at 520 nm.

### 2.11. Measurements of Protein and Lipid Oxidation of the Roasted Beef Patties

The beef powder was used to extract the total lipid. The samples’ extracted lipids were then weighed and kept in the dark at a temperature of −80 °C for later use. As per the literature, POV and TBARS were measured [16,17], and the results were expressed as mmol/kg lipid and mg MDA equivalents/kg lipid, respectively. Protein oxidation was reflected and evaluated by a protein carbonyl content detection kit, which adopted the method derivatization with dinitrophenylhydrazine (DNPH) [31].

### 2.12. Statistical Analysis

All samples were tested three times independently, and data were analyzed by Statistix 9.0 software (Tallahassee, FL, USA) using an analysis of variance to determine whether there were significant differences between treatments. For statistical significance, *p* < 0.05 was considered. The relative amount was calculated as follows: HAs or AGEs levels of beef patties treated with DES/HAs or AGEs levels of control.

## 3. Results and Discussion

### 3.1. TPC, TFC, and Antioxidant Capacity of Ginger Extracts

As a widely used spice and ingredient in China, ginger is rich in polyphenols and flavonoids. In this study (Figure 1A), the TPC in the ginger extract from conventional solvent ranged from 15.3 ± 1.10 to 29.6 ± 1.69 mg GAE/g DW, in which ginger extracted from water and 80% ethanol showed the highest and lowest TPC, respectively. However, compared with the conventional solvents, the TPC of the DESs extracts of ginger was significantly higher than that of the conventional solvent extract of ginger except for glycerol-based DESs (DES1, DES4, and DES 7). Among those DESs, DES2 showed that the highest TPC reached 50.6 ± 2.98 mg GAE/g DW, followed by DES 5, DES 6, and DES8. Moreover, the TFC of ginger extracts was also determined and showed similar results to that of TPC. As shown in Figure 1B, ginger extract from water and ethanol also showed the lowest TFC, which only reached 0.71 ± 0.13 mg RE/g DW and 3.04 ± 0.34 mg RE/g DW, respectively. Although the TFC of 80EtOH was significantly higher than that of the other two conventional solvents, it was still significantly lower than that of DESs, especially for DES2, DES5, DES6, and DES8, which ranged from 8.15 ± 0.59 to 9.80 ± 0.58 mg RE/g DW. In general, in comparison with different DES, lactic-acid-based DESs including DES 2, DES5, and DES8, especially for DES 2, showed significantly higher TPC and TFC than other DES. While glycerol-based DES including DES1, DES4, and DES 7, especially for DES1, showed a TPC and TFC comparable to conventional solvents and even slightly lower than 80EtOH, which is significantly lower than that of another DES. Our results are particularly consistent with earlier research, which found that the extraction yield of gingerols (an important phenolic compound from ginger) was significantly higher in 15 different DESs than in ethanol and water and that lactic-acid-based DESs also demonstrated the highest extraction yield of gingerols [15].

The ginger extract that was prepared under the ideal extraction conditions was then subjected to four antioxidant assays (DPPH, ABTS, FRAP, and ORAC) in the current study. According to earlier research, in vitro, antioxidant activity assays are frequently used to assess the efficacy of DES extracts since they are practical and simple to use [15,20,22]. As shown in Figure 2, the antioxidant activity of ginger extract with nine kinds of DESs was evaluated and compared with three kinds of conventional solvents (water, EtOH, and 80EtOH). As Figure 2A–C shows, the free radical scavenging activities of ginger extract from water were significantly (*p* < 0.05) lower than other solvents in the three antioxidant assays, which is consistent with our abovementioned results that ginger extract from water showed the lowest TPC and TFC. On the contrary, the ABTS, DPPH, and oxygen radical scavenging activities of lactic acid and xylitol-based DESs were significantly higher than that of conventional solvents, in which DES2 and DES 6 exhibited the highest DPPH and ABTS radical scavenging activities, followed by DES5, DES8, and DES9. Similarly, DES2, DES6, and DES8 also exhibited the highest oxygen radical scavenging activities, followed by DES5, DES8, and DES9. These results show that both lactic-acid- and xylitol-based DESs have excellent antioxidant activity. Despite that, the free radical scavenging activities of glycerol-based DESs (DES1, DES4, and DES 7) were the same as that of EtOH or even significantly lower than that of 80 EtOH. The respective range of DES1, DES4, and DES7 in the three antioxidant assays were 22.3 ± 2.52 to 32.7 ± 4.43 mg Trolox/g in the DPPH assay, 21.2 ± 1.47 to 41.2 ± 1.56 mg Trolox/g in the ABTS assay, and 66.3 ± 5.88 to 143 ± 6.03 mg Trolox/g in the ORAC assay.

The FRAP values of ginger extract show a similar trend to the free radical scavenging activities observed in the ABTS, DPPH, and ORAC tests for ginger extracted from various solvents. These results also showed that both lactic-acid- and xylitol-based DESs have excellent antioxidant activity. However, the FRAP value of DES2 was significantly higher than that of other DESs. The increased TPC and TFC of the DES2 compared to other DESs may explain its higher activity (Figure 1). Similar results showed that DES extracts had better antioxidant activity than those from using conventional solvents [19,20,21]. Although most of DESs are not thought to have antioxidant action [32]. However, studies have reported that some DES components can boost the antioxidant activity of extracts via a synergy between DES and soluble compounds [33].

### 3.2. Proximate and Texture Profile Analysis after Roasting

As Table 3 shown, the pH values, cooking loss, protein, and ash contents of the raw beef were 5.58 ± 0.13, 50.9% ± 1.22%, 42.9 ± 1.74 g/100 g, and 4.46% ± 0.08%, and these values were comparable with those reported in our previous studies for raw beef meat [12,14,16]. Raw beef samples were not measured for fat content since fat was removed before cooking in order to avoid its influence on the formation of HAs and AGEs. In each sample group, the cooking loss, protein, and ash content were comparable among the groups and showed a non-significant (*p* > 0.05) change between the control and patties treated with ginger extract. However, patties treated with lactic-acid-based DESs (DES2, DES4, and DES8), which varied from 4.61 ± 0.12 to 5.01 ± 0.19, were significantly (*p* < 0.05) lower than those of control patties.

The results of a textural profile analysis are shown in Table 4. The beef patties with or without addition of ginger extract showed non-significant (*p* > 0.05) difference concerning hardness, springiness, cohesiveness, gumminess, cohesiveness, and chewiness, which indicates that the different ginger extracts showed no significant (*p* > 0.05) change in the texture profile of beef patties.

### 3.3. Effects of Ginger Extract on the Formation of HAs in Beef Patties

As shown in Figure 3, six HAs (including PhIP, MeIQx, MeIQ, 4,8-MeIQx, Harmane, and Norharmane) were determined, and the effects of ginger extract on the formation of those HAs in the beef patties were reflected as results of the relative amount compared to the control group. Except for Harmane, which increased by 20%, water extract could lower HAs generation by 5–19%. Ginger extracted from the other two conventional solvents also reduced the formation of PhIP, MeIQx, MeIQ, and 4,8-MeIQx, in which MeIQ and 4,8-MeIQx were significantly decreased by 24–39% and 37–41%, respectively. This result is consistent with the study by Xue et al., who found that ginger can reduce the formation of HAs in a dose-dependent way by 0.5%, 1.0%, and 1.5%, inhibiting the synthesis of total HAs by 3.61%, 24.88%, and 27.42%, respectively [16]. On the contrary, Xue et al., indicated that the inhibition rates of 1.5% ginger on Norharmane and Harman were 28.56% and 27.92%, respectively [16]. However, EtOH and 80EtOH significantly increased the content of two β-carboline to 121% and 116% and 122% and 134%, respectively.

Moreover, all nine DES extracts considerably decreased the overall amount of HAs in beef patties. Of those DES extracts evaluated, the most effective in reducing the formation of HAs was DES2, which caused a signification reduction of 45% of the PhIP content, 30% of MeIQx, 51% of MeIQ, 79% of 4,8-MeIQx, 22% of Harmane, and 18% of Norharmane, and another two lactic-acid-based DESs showed similar inhibition effect on the formation of HAs. Roasting beef patties with xylitol-based DESs similarly resulted in much lower levels of the majority of HAs when compared to the control group except harmane, whose concentration rose to 109% in beef patties with DES9. For the glycerol-based DESs, the inhibition effects on HAs were significantly lower than other DESs, of which the relative amount of every HAs exhibited a similar change to that of conventional solvents. This result indicated that ginger extract exhibited inhibition ability to HAs, in which xylitol- and lactic-acid-based DESs showed better inhibition ability to HAs.

### 3.4. Effects of Ginger Extract on the Formation of CML and CEL in Beef Patties

According to Figure 4, the ginger extract had a significant inhibitory effect on CML and CEL. The water extract of ginger had the lowest inhibitory effect, and the content of CML and CEL was only reduced to 6.2% and 5.9%, respectively. This result is similar to the effects of ginger extract on the formation of HAs that were reported above. Another two ethanol extracts of ginger showed a better inhibition effect on the formation of CML and CEL, which decreased to 18.5–27.15% and 12.6–31.7%, respectively. Obviously, in comparison with conventional solvents, DES extracts of ginger exhibited better inhibitory capacity, which significantly decreased the formation of CML and CEL. For example, the inhibitory rate of DES2, DES5, and DES8 for bound CML in beef patties was 48.2%, 37.5%, and 41.9%, respectively, and it reached 58.5%, 52.7%, and 49.9% for bound CEL. Nonetheless, it is worth noting that glycerol-based DESs extract had a limited effect on the inhibition of CML and CEL, while the inhibitory rate of DES1, DES4, and DES7 for CML and CEL showed similar results with conventional solvent extracts.

Previous studies suggested that ginger and its main ingredient curcumin might be able to inhibit HAs, CML, and CEL by removing free radicals from the system and lowering the concentration of active carbonyl intermediates [16]. Therefore, the results of the inhibitory rate of HAs and AGEs in grilled beef patties by different ginger extracts are generally consistent with the results of the antioxidant activity of different ginger extracts. Moreover, the present study was further designed to prove this speculation by detecting the changes in the content of precursor and related oxidation products in beef patties.

### 3.5. Change of Glucose, Creatine, and Creatinine Contents

Creatine and creatinine are both important precursors that affect the formation of HAs [6]. Meanwhile, creatine and creatinine do not affect the synthesis of β-carboline HAs and AGEs, which are mostly produced from glucose and amino acids [6]. In addition to understanding the relationship between ginger extract and the development of HAs, AGEs, creatine, creatinine, and glucose, it was necessary to look into the precursors’ content changes [17,29]. In Table 5, precursor concentrations for creatine, creatinine, and glucose in roasted beef patties and beef patties treated with various ginger extracts are shown.

When compared to the control, those precursor levels in the water–extract-treated beef patties did not differ significantly. For EtOH and 80EtOH, the creatine level increased from 0.95 to 2.35 mg/g and 2.48 mg/g. In contrast, the batch of beef patties treated with DES extracts exhibited different phenomena, the creatine level increased from 0.95 to 1.33 mg/g and 5.97 mg/g, and both the creatinine and glucose levels ranged from 1.19 to 3.39 μmol/L and 0.17 to 0.40 mg/g, respectively. Compared to the control group and conventional solvent groups, roasted beef patties treated with DES2, DES5, DES6, and DES8 that exhibited the strongest inhibitory effect on Has and AGEs possess a significantly increasing trend of creatine, creatinine, and glucose. As a result, the inhibitory mechanism may have been caused by the DES extracts’ ability to impede the reactions of the precursors of glucose, creatine, and creatinine with some amino acids and intermediates. Previous studies also pointed out that the inhibitory effect of some bioactive compounds may be due to the hindering of pyridines and pyrazines via Strecker degradation, derived from glucose and amino acid [34,35].

### 3.6. Inhibitory Effects of Ginger Extract on Protein and Lipid Peroxidation

Lipids and proteins are susceptible to oxidative degradation during thermal processing, leading to the formation of reactive carbonyl compounds, which could further promote the accumulation of Has and AGEs [6,31]. Based on the results of some previous studies, we hypothesized and demonstrated that ginger extract significantly inhibits the formation of HAs and AGEs from roasted beef patties by reducing the amount of free radicals produced. As previously mentioned, ginger extracts demonstrated excellent antioxidant activity. Therefore, the effect of ginger extract on lipid and protein oxidation was determined.

Protein carbonyls, which are produced during protein oxidative breakdown, show the degree of protein oxidation, while lipid oxidation is commonly measured with TBARS and POV values [31]. As shown in Figure 5, total carbonyl content, TBARS, and POV values in the control group were 5.90 ± 0.90 mmol/kg, 15.2 ± 1.13 mg MDA eq/kg lipid, and 36.7 ± 1.51 mmol/kg, respectively. All of the ginger extracts could significantly decrease the total carbonyl content, TBARS, and POV values in beef patties, in which total carbonyl content, TBARS, and POV values decreased to 3.30–5.30 mmol/kg, 11.3–14.7 mg MDA eq/kg lipid, and 22.0–32.8 mmol/kg in the conventional solvent groups, respectively. Compared with conventional solvent groups, DES1, DES3, DES4, DES7, DES8, and DES9 exhibited similar effects on protein and lipid peroxidation, but DES2, DES5, and DES6 could significantly reduce the oxidative degradation of lipid and protein in roasted beef patties, which is consistent with the results of the TPC, TFC, and in vitro antioxidant capacity of ginger extracts.

Overall, our results imply that ginger extract reduces lipid and protein peroxidation by quenching free radicals, as described above. Future experiments should investigate changes in active carbonyl intermediates.

## 4. Conclusions

Generally, HAs and AGEs are often produced simultaneously during the thermal processing of food. Despite this, there have been few studies exploring simultaneous inhibition methods for these two harmful compounds, which certainly cannot satisfy public demands for food safety. In the present study, ginger extracts with strong antioxidant activity were prepared with various deep eutectic and conventional solvents. Our results further indicate the use of ginger DES extracts in the preparation of beef patties as a way to lessen the generation of HAs and AGEs during roasting. As a whole, all ginger extract treatments significantly reduced the amount of HAs and AGEs to 18–79% and 41–58%, respectively. It appears that xylitol- and lactic-acid-based DES extracts reduce the amount of HAs and AGEs to the greatest extent possible. Further, we evaluated the physical/chemical changes of ginger extracts on the beef patties and clarified that inhibiting ability of ginger extracts for HAs and AGEs mainly originated from quenching of free radicals through several studies including antioxidant index, precursors of HAs and AGEs, and lipid and protein oxidation. The present study provides a theoretical framework for understanding and developing the efficient inhibition of HAs and AGEs development in thermally processed foods based on DES-prepared natural product extracts. It may be relevant to different meat systems and useful to food producers for producing beef patties with less HA and AGE formation. This has significant implications for developing healthier meat products and improving food safety.

## Figures and Tables

**Figure 1 foods-11-03161-f001:**
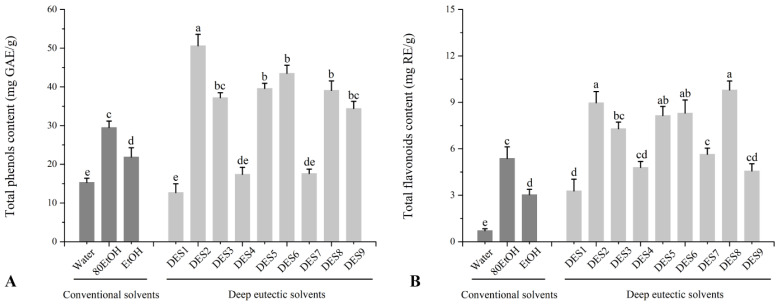
Effect of extraction solvent (conventional and deep eutectic solvents) on (**A**) total phenolic and (**B**) flavonoids contents of ginger extract. The corresponding to the solvent’s abbreviations can be consulted in Table 1. Values are expressed as mean ± SD (*n* = 3). Different letters in each series indicate significant differences (*p* < 0.05); lowercase letters denote significant differences (*p* < 0.05) between solvents.

**Figure 2 foods-11-03161-f002:**
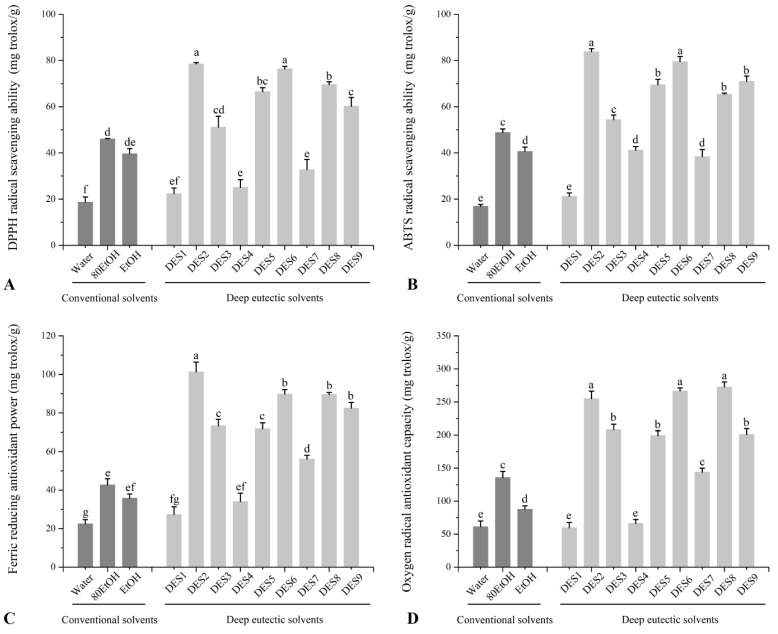
Effect of extraction solvent (conventional and deep eutectic solvents) on antioxidant capacity of ginger extract determined by (**A**) DPPH, (**B**) ABTS, (**C**) FRAP, and (**D**) ORAC assays. The corresponding solvents’ abbreviations can be consulted in Table 1. Values are expressed as mean ± SD (*n* = 3). Different letters in each series indicate significant differences (*p* < 0.05); lowercase letters denote significant differences (*p* < 0.05) between solvents.

**Figure 3 foods-11-03161-f003:**
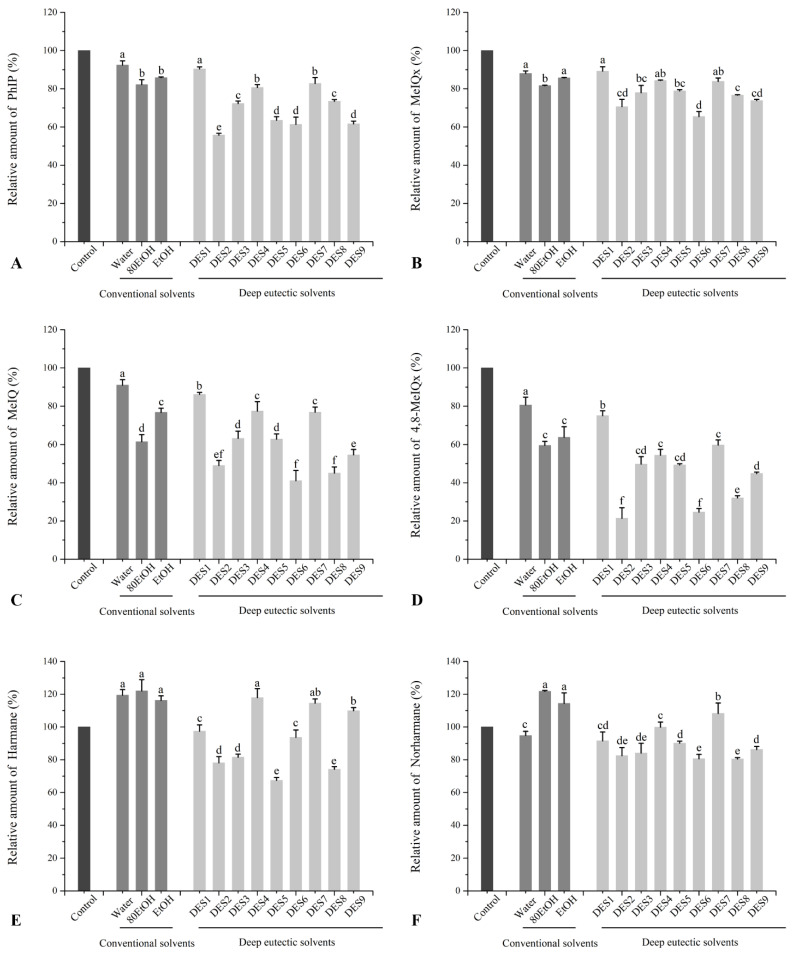
Relative amounts of (**A**) PhIP, (**B**) MeIQx, (**C**) MeIQ, (**D**) 4,8-DiMeIQx, (**E**) Harmane, and (**F**) Norharmane in roasted beef patties with different ginger extracts. The corresponding solvents’ abbreviations can be consulted in Table 1. Values are expressed as mean ± SD (*n* = 3). Different letters in each series indicate significant differences (*p* < 0.05); lowercase letters denote significant differences (*p* < 0.05) between solvents.

**Figure 4 foods-11-03161-f004:**
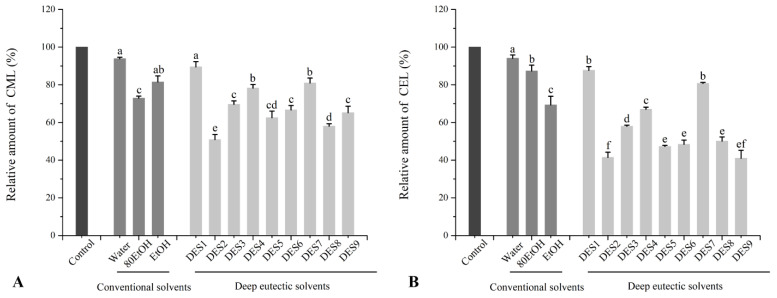
Relative amounts of (**A**) CML and (**B**) CEL in roasted beef patties with different ginger extracts. The corresponding to the solvent’s abbreviations can be consulted in Table 1. Values are expressed as mean ± SD (*n* = 3). Different letters in each series indicate significant differences (*p* < 0.05); lowercase letters denote significant differences (*p* < 0.05) between solvents.

**Figure 5 foods-11-03161-f005:**
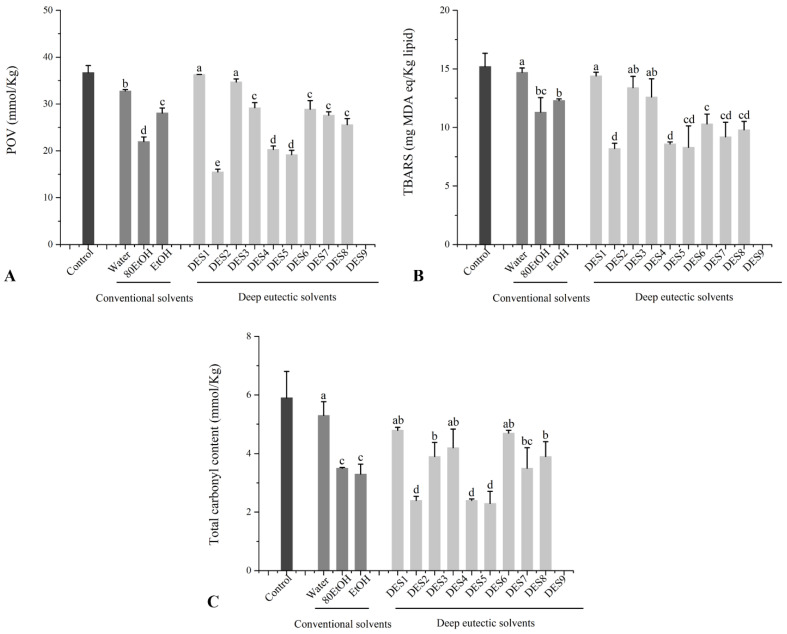
Effect of different ginger extract on lipid and protein oxidation of beef patties determined by (**A**) POV, (**B**) TBARS, and (**C**) total carbonyl content. The corresponding solvents’ abbreviations can be consulted in Table 1. Values are expressed as mean ± SD (*n* = 3). Different letters in each series indicate significant differences (*p* < 0.05); lowercase letters denote significant differences (*p* < 0.05) between solvents.

**Table 1 foods-11-03161-t001:** The composition of nine kinds of deep eutectic solvents.

Name of DES	Component 1 (HBA)	Component 2 (HBD)	Molar Ratio
DES1	Choline chloride	Glycerol	1:2
DES2	Choline chloride	Lactic acid	1:2
DES3	Choline chloride	Xylitol	2:1
DES4	Betaine	Glycerol	1:2
DES5	Betaine	Lactic acid	1:2
DES6	Betaine	Xylitol	2:1
DES7	L-carnitine	Glycerol	1:2
DES8	L-carnitine	Lactic acid	1:2
DES9	L-carnitine	Xylitol	2:1

**Table 2 foods-11-03161-t002:** Description of the MRM parameters for the analysis of HAs and AGEs in beef patties.

	Precursor Ion (*m*/*z*)	Product Ion (*m*/*z*)	Cone Voltage (V)	Collision Voltage (eV)	Dwell Time (s)
HAs					
DMIP	163	148	30	25	0.10
Phe-p-1	171	127	30	35	0.10
1,5,6-TMIP	177	162	30	25	0.15
Harmane	183	115	30	30	0.15
Norharmane	169	115	30	35	0.10
AαC	183	140	30	35	0.15
MeAαC	198	181	30	30	0.15
Glu-p-1	199	145	30	35	0.10
IQ	199	130	30	35	0.15
IQ [4,5-b]	199	115	30	35	0.10
IQx	200	185	30	35	0.10
MeIQ	213	198	30	30	0.10
MeIQx	214	131	30	35	0.10
PhIP	225	210	30	35	0.15
4,8-DiMeIQx	228	212	30	30	0.15
7,8-DiMeIQx	228	213	30	35	0.15
4,7,8-DiMeIQx	242	227	30	30	0.10
AGEs					
CML	205	84	20	18	0.15
*d_4_*-CML	209	88	20	18	0.15
CEL	219	84	22	20	0.15
*d_4_*-CEL	223	88	22	20	0.15

**Table 3 foods-11-03161-t003:** Chemical composition, cooking loss, and pH of the beef patties with different ginger extracts.

Group	pH	Protein (g/100g)	Cooking Loss (%)	Ash (%)
Control	5.58 ± 0.13 a	42.9 ± 1.74 ab	50.9 ± 1.22 a	4.46 ± 0.08 a
Water	5.77 ± 0.09 a	43.4 ± 0.57 a	48.3 ± 2.96 a	4.31 ± 0.01 a
80EtOH	5.74 ± 0.07 a	45.1 ± 2.56 a	52.3 ± 1.05 a	4.16 ± 0.09 a
EtOH	5.62 ± 0.04 a	44.3 ± 2.53 a	49.9 ± 2.90 a	4.53 ± 0.03 a
DES1	5.69 ± 0.05 a	46.2 ± 2.98 a	50.2 ± 1.17 a	4.27 ± 0.05 a
DES2	5.01 ± 0.19 b	44.0 ± 1.94 a	52.3 ± 1.08 a	4.33 ± 0.12 a
DES3	5.73 ± 0.16 a	45.2 ± 2.38 a	49.9 ± 2.46 a	4.86 ± 0.04 a
DES4	5.81 ± 0.03 a	46.7 ± 1.08 a	52.9 ± 1.38 a	4.88 ± 0.20 a
DES5	4.61 ± 0.12 b	46.2 ± 2.19 a	48.0 ± 1.80 a	4.85 ± 0.07 a
DES6	5.88 ± 0.05 a	46.9 ± 0.67 a	52.5 ± 1.21 a	4.45 ± 0.13 a
DES7	5.33 ± 0.02 a	43.1 ± 1.04 ab	48.7 ± 2.10 a	4.55 ± 0.04 a
DES8	4.67 ± 0.09 b	47.7 ± 1.96 a	48.1 ± 2.89 a	4.79 ± 0.05 a
DES9	5.78 ± 0.17 a	45.9 ± 1.26 a	52.7 ± 1.36 a	4.86 ± 0.07 a

Mean ± SD (*n* = 3). Means with different letters (a, b) in the same column are significantly different (*p* < 0.05).

**Table 4 foods-11-03161-t004:** Texture characteristics of raw beef and roast beef patties with different ginger extracts.

Group	Hardness (*N*)	Springiness (mm)	Gumminess (*N*)	Cohesiveness (*N*)	Chewiness (mJ)
Control	8298 ± 51 ab	0.39 ± 0.02 a	4954 ± 57 a	0.65 ± 0.15 ab	3728 ± 52 a
Water	8276 ± 43 b	0.49 ± 0.08 a	4832 ± 83 a	0.91 ± 0.14 a	3807 ± 77 a
80EtOH	8294 ± 59 ab	0.52 ± 0.07 a	4905 ± 30 a	0.96 ± 0.16 a	3722 ± 73 a
EtOH	8328 ± 96 a	0.32 ± 0.03 a	4859 ± 79 a	0.83 ± 0.19 a	3698 ± 49 ab
DES1	8394 ± 59 a	0.44 ± 0.07 a	4858 ± 96 a	0.73 ± 0.05 a	3672 ± 65 ab
DES2	8393 ± 103 a	0.36 ± 0.02 a	4901 ± 44 a	0.69 ± 0.11 a	3634 ± 62 ab
DES3	8478 ± 101 a	0.46 ± 0.06 a	4961 ± 48 a	0.57 ± 0.17 ab	3729 ± 65 a
DES4	8306 ± 83 ab	0.4 ± 0.07 a	4834 ± 110 a	0.74 ± 0.04 a	3799 ± 29 a
DES5	8344 ± 50 a	0.5 ± 0.02 a	4924 ± 34 a	0.78 ± 0.02 a	3710 ± 88 a
DES6	8437 ± 99 a	0.59 ± 0.10 a	4927 ± 103 a	0.75 ± 0.12 a	3842 ± 34 a
DES7	8257 ± 67 ab	0.52 ± 0.01 a	4841 ± 101 a	0.71 ± 0.01 a	3654 ± 72 ab
DES8	8455 ± 60 a	0.57 ± 0.03 a	4801 ± 98 a	0.66 ± 0.16 a	3628 ± 64 ab
DES9	8240 ± 96 ab	0.51 ± 0.06 a	4832 ± 41 a	0.60 ± 0.03 b	3733 ± 21 a

Mean ± SD (*n* = 3). Means with different letters (a, b) in the same column are significantly different (*p* < 0.05).

**Table 5 foods-11-03161-t005:** Contents of precursors creatine, creatinine, and glucose in beef patties with different ginger extract.

Group	Creatine (mg/g)	Creatinine (μmol/L)	Glucose (mg/g)
Control	0.95 ± 0.04 f	1.22 ± 0.08 cd	0.11 ± 0.01 d
Water	1.21 ± 0.03 e	1.55 ± 0.19 c	0.13 ± 0.01 d
80EtOH	2.35 ± 0.07 c	1.85 ± 0.16 bc	0.21 ± 0.03 bc
EtOH	2.48 ± 0.26 c	2.45 ± 0.21 b	0.12 ± 0.01 d
DES1	1.33 ± 0.29 ef	1.19 ± 0.05 cd	0.17 ± 0.00 b
DES2	5.30 ± 0.20 a	3.26 ± 0.02 a	0.40 ± 0.08 a
DES3	2.72 ± 0.02 c	2.19 ± 0.16 b	0.26 ± 0.07 a
DES4	1.48 ± 0.28 d	2.52 ± 0.11 b	0.22 ± 0.03 b
DES5	4.80 ± 0.27 a	3.34 ± 0.21 a	0.36 ± 0.06 a
DES6	5.97 ± 0.08 a	2.54 ± 0.12 b	0.31 ± 0.09 a
DES7	1.89 ± 0.11 d	2.30 ± 0.05 b	0.27 ± 0.00 b
DES8	4.22 ± 0.04 b	3.39 ± 0.02 a	0.28 ± 0.07 a
DES9	4.44 ± 0.13 b	2.47 ± 0.03 b	0.29 ± 0.05 a

Mean ± SD (*n* = 3). Means with different letters (a–f) in the same column are significantly different (*p* < 0.05).

## Data Availability

The data presented in this study are available on request from the corresponding author.

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
