# Peer review of "The Influence of Deep Eutectic Solvents Extract from Ginger on the Formation of Heterocyclic Amines and Advanced Glycation End Products in Roast Beef Patties"

_foods, 2022, doi:10.3390/foods11203161_

Round 1
Reviewer 1 Report
This manuscript presents an original study about the influence of deep eutectic solvents extract from ginger on the formation of heterocyclic amines and advanced glycation end products in roast beef patties.
The experimental data provided a theoretical framework for understanding ginger extract’s ability to suppress HA and AGE development in heat-processed roast beef patties.
The scientific quality of the manuscript it rises to the scientific level of the Food Journal. The technical quality of the manuscript is satisfactory in terms of how it was written and how the experimental results are presented. The style of expression reflects the scientific training of the authors. The manuscript is partially edited in accordance with the article drafting requirements.
The Abstract is concise and contains sufficient information to highlight the content of the article and the Introduction section provides a clear statement of the problem studied in the present manuscript.
The Materials and Methods section is satisfactorily presented and appropriate to the purpose of the research.
Results follow the guidelines described in the Author's Guide and are poorly discussed. Bibliographic references of authors who have results in this field are not cited.
References are relevant and current and follow the journal’s format.
Regarding English Proficiency: there are a few minor language corrections to be made. The authors are advised to carefully check the entire manuscript.
The Conclusions of the article partially reflect the results of the given study.
I also suggest the following:
Please edit the manuscript according to the requirements of Foods, especially the citation of bibliographic references.
Please correct chemistry formulas (Na2CO3 and not Na2CO3, etc.).
Not all abbreviations are explained in the basic text (DPPH, ABTS; FRAP, TPA, etc.).
Line 112. Indicate the concentration of EtOH and at 80 EtOH must be added (v/v).
Line 128. It is not clear why "According to Quan et al, 25 After mixing 20 μL..." in this phrase "After" is written with a capital letter. There are other cases. Review the entire manuscript.
Line 194. Research methods are not clearly described. Also, check the citations to bibliographic references in this section.
From the Materials and methods section, it is not clear how the control group was formed.
Line 207. Please indicate the software used for statistical processing.
In section 3.1. there are no citations to bibliographical references of other authors in the field of the manuscript.
In section 3.2. there are no citations to bibliographic references that confirm the results obtained by the authors. There is the phrase "these values were comparable to those reported in the literature for raw beef" but there are no references to bibliographic references that confirm this statement.
Lines 290-295. Texture characteristics are not explained.
Change the name of Table 3, because cooking losses are not part of the chemical composition.
From Table 4 it is not clear which results are attributed to raw beef and roast beef patties.
Lines 369-370. Check the measurement unit for creatinine content (μmol/L or mg/g). Also in table 5.
Lines 367, 369, 370, etc. I suggest you do not show the value of the standard deviation in the basic text (ex. 0.95 to 2.35 mg/g and 2.48 mg/g).
Check the units of measure in figure 5 (y-axis). Also, the units of measure of the TBARS and POV parameters in the basic text do not correspond to figure 5.
Author Response
To Referee 1:
Q: Please edit the manuscript according to the requirements of Foods, especially the citation of bibliographic references.
R: Thank you very much. I have checked and edited the manuscript according to the guideline of Foods, and cited appropriate references in the main text (see Line 245-249, 257-258, 286-287, 293-294, and 387-390 in revised manuscript).
Q: Please correct chemistry formulas (Na2CO3 and not Na2CO3, etc.).
R: Thank you very much. I have corrected chemistry formulas in the text (see Line 121, 127-128, and 192 in revised manuscript).
Q: Not all abbreviations are explained in the basic text (DPPH, ABTS; FRAP, TPA, etc.).
R: Thank you very much. I have added the full name of those abbreviations in the main text (see Line 13, 23-24, 97-101, 142-143, 146, and 170 in revised manuscript).
Q: Line 112. Indicate the concentration of EtOH and at 80 EtOH must be added (v/v).
R: Thank you very much. I have revised this sentence according to your suggestion (see Line 117 in revised manuscript).
Q: Line 128. It is not clear why "According to Quan et al, 25 After mixing 20 μL..." in this phrase "After" is written with a capital letter. There are other cases. Review the entire manuscript.
R: Thank you very much. I feel sorry for my mistakes, and I have revised those sentences (see Line 134 in revised manuscript).
Q: Line 194. Research methods are not clearly described. Also, check the citations to bibliographic references in this section.
R: Thank you very much. I have revised this section, and described the method adopted in this section, since the content of glucose and creatinine was measured by commercial kits, and all the experimental operations were performed according to the instructions of commercial kits, so the specific experimental operation process is not described in detail (see Line 203-208 in revised manuscript).
Q: From the Materials and methods section, it is not clear how the control group was formed.
R: Thank you very much. I feel sorry for my mistake, since I did not provide such important information, and I have defined the control group in section 2.6 (see Line 155-156, 159-162 in revised manuscript).
Q: Line 207. Please indicate the software used for statistical processing.
R: Thank you very much. I have added the software used for statistical processing (see Line 220-221 in revised manuscript).
Q: In section 3.1. there are no citations to bibliographical references of other authors in the field of the manuscript.
R: Thank you very much. I have cited some appropriate references in this section (see Line 245-249, 257-258, and 286-287 in revised manuscript).
Q: In section 3.2. there are no citations to bibliographic references that confirm the results obtained by the authors. There is the phrase "these values were comparable to those reported in the literature for raw beef" but there are no references to bibliographic references that confirm this statement.
R: Thank you very much. I feel sorry for my mistakes, and I have cited the reference to confirm the results obtained by the authors (see Line 293-294 in revised manuscript).
Q: Lines 290-295. Texture characteristics are not explained.
R: Thank you very much. I have checked the results of texture profile analysis and revised those sentences, since there were non-significantly difference between different experimental groups (see Line 302-306 in revised manuscript).
Q: Change the name of Table 3, because cooking losses are not part of the chemical composition.
R: Thank you very much. I completely agree with your suggestion, and I have revised the name of table 3 (see Line 307-308 in revised manuscript).
Q: From Table 4 it is not clear which results are attributed to raw beef and roast beef patties.
R: Thank you very much. I feel sorry that my previous description was wrong and caused you a lot of confusion, and I have revised those sentences in section 3.2 (see Line 302-306 in revised manuscript).
Q: Lines 369-370. Check the measurement unit for creatinine content (μmol/L or mg/g). Also in table 5.
R: Thank you very much. I feel sorry for my mistakes, the unit for creatinine content should be μmol/L, and I have revised it (see Line 378-381 in revised manuscript).
Q: Lines 367, 369, 370, etc. I suggest you do not show the value of the standard deviation in the basic text (ex. 0.95 to 2.35 mg/g and 2.48 mg/g).
R: Thank you very much. I completely agree with your suggestion, and I have revised those sentences (see Line 378-381 in revised manuscript).
Q: Check the units of measure in figure 5 (y-axis). Also, the units of measure of the TBARS and POV parameters in the basic text do not correspond to figure 5.
R: Thank you very much. The units of measure in figure 5 are right, I have revised the wrong units of measure of the TBARS and POV in the main text (see Line 402-407 in revised manuscript).

Reviewer 2 Report
1) Add nomenclature to the manuscript
2) Please highlight the novelty of this study in your manuscript
3) What about the experimental conditions of all figures from 1 to 5 and table from 1-5
4) Compare and thoroughly discuss your results with the literature
5) In table 4. Texture characteristics of raw beef and roast beef patties with different ginger extract (column Hardness (N)), please check these values ±51ab, ±43b, ±59ab……..,±99a….., the average is 8!!!
6) Quality of all figures is poor
7) Please check and add the unit of the value in all tables (Unit of protein in table 3 is missing for example)
8) The conclusion should be more quantitative
9) Literature review should be updated with a new reference related to the influence of deep eutectic solvents extract from ginger on the formation of heterocyclic amines and advanced glycation end products in roast beef patties
10) There are some typographical and grammatical errors in the manuscript. Hence, the manuscript should be carefully checked, and necessary corrections should be done
Author Response
To Referee 2:
Q: Add nomenclature to the manuscript
R: Thank you very much. I have added the definition of control and experiment groups in section 2.6 (see Line 159-162 in revised manuscript).
Q: Please highlight the novelty of this study in your manuscript
R: Thank you so much. according to your suggestion, I have revised some sentences in abstract, introduction and conclusion to highlight the novelty of our present study (see Line 13-16, 27-29, 86-91, and 434-440in revised manuscript).
Q: What about the experimental conditions of all figures from 1 to 5 and table from 1-5
R: Thank you for your good suggestion. The experimental conditions of all the results have been described in section 2, and all the experiments were tested for three times and data were analyzed by analysis of variance (see Line 155-156, 159-162, 204-204, and 216-217 in revised manuscript).
Q: Compare and thoroughly discuss your results with the literature
R: Thank you so much for your good suggestion. I have compared and discussed the results with some previous literatures and cited appropriate references (see Line 245-249, 257-258, 286-287, 293-294, and 387-390 in revised manuscript).
Q: In table 4. Texture characteristics of raw beef and roast beef patties with different ginger extract (column Hardness (N)), please check these values ±51ab, ±43b, ±59ab……..,±99a….., the average is 8!!!
R: Thank you so much for your kind advice. I feel sorry but the average value of texture characteristics in eight thousand. Moreover, I have checked the results of texture profile analysis and revised those sentences, since there were non-significantly difference between different experimental groups (see Line 302-306 in revised manuscript).
Q: Quality of all figures is poor
R: Thank you so much for your kind advice, I have rework figure 3, and the DPI of other figures have been modified (see Line 336 and other figures in revised manuscript).
Q: Please check and add the unit of the value in all tables (Unit of protein in table 3 is missing for example)
R: Thank you so much for your kind advice. I feel sorry that there were some mistakes about the units of the value in the tables and main text, and I have revised those sentences (see Line 308, 378-382 in revised manuscript).
Q: The conclusion should be more quantitative
R: Thank you so much for your good suggestion. I have revised some sentence in the conclusion (see Line 428-430 in revised manuscript).
Q: Literature review should be updated with a new reference related to the influence of deep eutectic solvents extract from ginger on the formation of heterocyclic amines and advanced glycation end products in roast beef patties
R: Thank you very much for your suggestion. I have cited our previously publish reference that reported the inhibition effect of ginger on the formation of heterocyclic amines and advanced glycation end products in roast beef patties. However, we did not search for the latest research reports on the inhibition of heterocyclic amines and advanced glycation products in barbecued meat by deep eutectic solvent extracts.
Here are the reference:
- Xue, C., Deng, P., Quan, W., Li, Y., He, Z., Qin, F., ... & Zeng, M. (2022). Ginger and curcumin can inhibit heterocyclic amines and advanced glycation end products in roast beef patties by quenching free radicals as revealed by electron paramagnetic resonance. Food Control, 138, 109038.
Q: There are some typographical and grammatical errors in the manuscript. Hence, the manuscript should be carefully checked, and necessary corrections should be done
R: Thank you so much for your reviewing and giving good suggestions to me. According to your suggestion, this manuscript has been revised by editors whose native language is English for language corrections. Since we considered that we worked on the manuscript for a long time and the repeated addition and removal of sentences and sections obviously led to poor readability.
Here are the certificate of editing:

Round 2
Reviewer 1 Report
Dear Authors,
Please edit the manuscript according to the requirements of Foods, especially the citation of bibliographic references (e.g. [1]).
Line 166. Exclude "...fat, ...., and moisture contents was analyzed". These are not presented in the basic text.
Author Response
To Referee 1:
Q: Please edit the manuscript according to the requirements of Foods, especially the citation of bibliographic references (e.g. [1]).
R: Thank you for your patient review, I feel sorry that in my last revised manuscript I did not noticed the problems about the citation of bibliographic references, but thanks for your kind reminder, I have revised those problems in the main text and references (see Line 35, 37, 41, 46, 48, 52, 54, 56, 57, 62, 65, 69, 73, 78, 80, 84, 107, 116, 121, 127, 135, 140, 144, 148, 158, 165, 171, 175, 178, 191, 197, 203, 206, 208, 217, 220, 260, 289-292, 296, 322, 359, 371, 372, 375, 392, 396, and 403 in revised manuscript).
Q: Line 166. Exclude "...fat, ...., and moisture contents was analyzed". These are not presented in the basic text.
R: Thank you very much. I have revised this sentences according to your suggestion (see Line 166 in revised manuscript).

Reviewer 2 Report
No response to my comment (reviewer 2) Add nomenclature to the manuscript. The same comment asked by the First reviewer (Not all abbreviations are explained in the basic text (DPPH, ABTS; FRAP, TPA, etc.)
Author Response
Q: No response to my comment (reviewer 2) Add nomenclature to the manuscript. The same comment asked by the First reviewer (Not all abbreviations are explained in the basic text (DPPH, ABTS; FRAP, TPA, etc.).
R: Thank you for you kind advice. I'm sorry for not giving a correct response to your valuable comments in my last revision. Because I misunderstood that you meant adding nomenclature to the manuscript is adding a description of the sample name (e.g. DES1-9 EtoH and 80EtoH). Now I guess you are referring to the nomenclature for abbreviations in the manuscript, although I have added the full name of the abbreviation where it first appears in the text, but according to your suggestion, I have listed all the abbreviations individually at the end of the manuscript (see Line 452-461 in revised manuscript).
